# Thermodynamic Analysis to Evaluate the Effect of Diet on Brain Glucose Metabolism: The Case of Fish Oil

**DOI:** 10.3390/nu16050631

**Published:** 2024-02-24

**Authors:** Cennet Yildiz, Isabel Medina

**Affiliations:** 1Marine Chemistry, Instituto de Investigaciones Marinas CSIC, 36208 Vigo, Spain; medina@iim.csic.es; 2Biothermodynamics, School of Life Sciences, Technische Universität München, 85354 Freising, Germany

**Keywords:** brain glucose metabolism, n-3 PUFAs, fish oil supplementation, brain ageing, entropy, Gibbs free energy

## Abstract

Inefficient glucose metabolism and decreased ATP production in the brain are linked to ageing, cognitive decline, and neurodegenerative diseases (NDDs). This study employed thermodynamic analysis to assess the effect of fish oil supplementation on glucose metabolism in ageing brains. Data from previous studies on glucose metabolism in the aged human brain and grey mouse lemur brains were examined. The results demonstrated that Omega-3 fish oil supplementation in grey mouse lemurs increased entropy generation and decreased Gibbs free energy across all brain regions. Specifically, there was a 47.4% increase in entropy generation and a 47.4 decrease in Gibbs free energy in the whole brain, indicating improved metabolic efficiency. In the human model, looking at the specific brain regions, supplementation with Omega-3 polyunsaturated fatty acids (n-3 PUFAs) reduced the entropy generation difference between elderly and young individuals in the cerebellum and particular parts of the brain cortex, namely the anterior cingulate and occipital lobe, with 100%, 14.29%, and 20% reductions, respectively. The Gibbs free energy difference was reduced only in the anterior cingulate by 60.64%. This research underscores that the application of thermodynamics is a comparable and powerful tool in comprehending the dynamics and metabolic intricacies within the brain.

## 1. Introduction

The human brain is an exceedingly complex and sophisticated organ characterized by a remarkable capacity for processing and integrating vast amounts of information. The human brain represents a mere 2% of our total body mass, yet it consumes about 20% of the whole body’s oxygen and 25% of its resting glucose. Glucose is the primary energy substrate utilized by the brain, with few exceptions. Interestingly, studies using arterio-venous catheterization during stress-induced fasting have shown that the brain switches from primarily utilizing glucose to more than 50–60% ketone bodies [1]. Neurons and astrocytes require a continual supply of glucose and oxygen to maintain proper functioning. Neurons, being the most significant energy consumers among brain cells, account for 70% to 80% of overall brain energy expenditure, with the remaining portion utilized by glial cells, including astrocytes, oligodendrocytes, and microglial cells [2]. Lactate oxidation increases during neural activity. When astrocytes reabsorb glutamate, it activates glucose utilization and lactate production and release to sustain neural energy needs as activation continues [3]. When glucose is processed in the brain through glycolysis, the TCA cycle, and oxidative phosphorylation, it is almost entirely converted to CO_2_ and water, producing between 30 and 36 ATPs per molecule of glucose depending on the efficiency of oxidative phosphorylation [4,5,6,7], as schematically represented in Figure 1. However, cerebral glucose metabolic rate decreases with age for the whole brain and the temporal, parietal, temporal, and occipital lobes [8,9,10,11]. Inefficiency in metabolizing glucose during ageing is associated with damage in glial cells and neurons, thereby influencing the pathways for glucose transport, glycolytic and Krebs cycle enzyme activities, and insulin signalling. However, attempts to associate ageing and metabolic variables with glucose metabolism have been inconclusive, leaving many uncertainties. In addition, the complex neuroenergetic processes in the brain require advanced techniques such as neuroimaging, allowing for quantitative and non-invasive assessments of brain energy metabolism and the quantification of metabolic rates.

Thermodynamic analysis has emerged as a powerful tool for the quantification of brain function and the brain’s information processing capacity. In thermodynamics, the brain is viewed as a complex system that interacts with its surroundings, allowing energy, entropy, and information to flow through its boundaries. Various thermodynamic variables, such as energy, temperature, and pressure, influence its cognitive functions. The brain is considered an isothermal and isobaric system, and its thermodynamic calculations are based on its chemical nourishment through the blood vessels. The brain consumes more energy and generates more entropy during states of awareness and thinking. In this field, the use of entropy is leading to noteworthy research in studies on the ageing brain and the quantification of the information processing of brain networks [12]. Entropy, as a tool used to quantify brain complexity, is crucial in the adaptation processes, and the loss of such complexity due to ageing increases one’s susceptibility to diseases and disorders. Brain regions with high metabolic activity in healthy individuals tend to generate more entropy [13]. A recent study by Yildiz and Özilgen (2022) [13] discussed the age dependency of cerebral metabolic (CMRGlc) Gibbs free energy utilization rates and related entropy generation rates, along with the damage associated with the ageing process. Indeed, entropy is produced as a result of metabolic activity in biological systems; most is removed, and only a small fraction accumulates. Entropy accumulation has been revealed to be a structural deterioration and is considered a sign of ageing [14,15,16,17]. Thus, the use of entropy generation has yielded auspicious results for substantially advancing the implementation of precision lifespan gerontological predictions [18].

The concept of entropy generation has been applied to comprehend the influence of diet composition on human ageing [19]. The first and second laws of thermodynamics have been proposed as a basis for predicting the impact of nutrition on the human lifespan [18]. There has been a substantial increase in evidence demonstrating a connection between the role of diet and nutrition and ageing, as well as age-related cognitive decline. As individuals progress in age, there is an increased vulnerability to cognitive decline and reduced cerebral glucose metabolism. 

Ensuring sufficient dietary intake of omega-3 polyunsaturated fatty acids (n-3 PUFAs) is widely recognized to foster optimal cognitive function during the ageing process [20]. Research has revealed that incorporating marine n-3 PUFA into the diet can yield favourable results on behaviour, mood, and specific cognitive conditions [21,22,23]. As a considerable number of individuals lack EPA and DHA, the consumption of marine food and marine omega-3 PUFA supplements provides an affordable means to promote mental and emotional well-being; this has led to them being incorporated into daily regimens [24,25]. A lack of dietary n-3 PUFAs and decreased brain DHA content have been associated with severe changes in neurotransmission processes, which demand high levels of brain ATP [25]. Thus, impaired brain glucose utilization and an n-3 PUFA-deficient diet have been studied together in an attempt to identify alternatives that guarantee a healthy aged brain [20]. 

This study aimed to assess the ability of thermodynamic measures based on Gibbs free energy utilization rates and related entropy generation rates to describe the effect supplementation with fish oil and n-3 PUFAs has on glucose metabolism in the ageing brain. To do this, ^18^F-fluorodeoxyglucose (^18^F-FDG-PET) data available for an animal model of grey mouse lemurs fed marine oils for 12 months were analysed [26], and previously published ^18^F-FDG PET data corresponding to an interventional study in aged humans fed marine oils for 3 weeks were also analysed [27]. For both models, specific brain areas were studied. The functions of these areas are presented in Table A1 to illustrate why they were selected for both models.

## 2. Materials and Methods

### 2.1. Thermodynamic Assessment of Brain Glucose Metabolism

The brain is an open system where glucose and O_2_ are supplied through blood flow. In the brain of a healthy and dynamic individual, the inflow and outflow of blood are balanced [28]. Figure 2 illustrates a healthy brain as a thermodynamic system. With sufficient oxygen, glucose transforms carbon dioxide and metabolic water through oxidative phosphorylation. Conversely, under inadequate oxygen conditions, glycolysis converts glucose into lactic acid with glycolysis. The glucose oxidation reaction is presented in Equation (1).
C_6_H_12_O_6(s)_ + 6O_2(g)_ ⟶ 6H_2_O_(g)_ + 6CO_2(g)_(1)

Applying this equation to the brain assumes that glucose and oxygen enter the body at 25 °C, while carbon dioxide and metabolic water exit at 37 °C. The heat generation rate due to metabolism (*Q_m_*) was calculated using Equation (2):(2)$$Q_{m}=\sum n_{p}{\left(h_{f}^{-o}+h^{-}-h^{-o}\right)}_{p}-\sum n_{r}\;{\left(h_{f}^{-o}+h^{-}-h^{-o}\right)}_{r}$$
where $n_{p}$ and $n_{r}$ indicate the mole number of products and reactants, and $h_{f}^{-o}$, $h^{-}$, and $h^{-o}$ denote the formation of enthalpies at the standard conditions, respectively. The thermodynamic properties of the chemicals are presented in Table 1.

By using Equation (3), the metabolic work performance of the brain was calculated.
(3)$${ŋ}_{ATP}=\frac{W_{ATP}}{Q_{m}}=\frac{Total\;work\;obtained\;from\;ATp\;molecules}{Total\;heat\;production}\;$$

In Equation (3), ${ŋ}_{ATP}$ signifies the brain’s metabolic efficiency, which is described as the ratio of total work from ATP molecules to heat production, and $W_{ATP}$ represents the work performance from heat energy. $W_{ATP}$ is determined by considering ${ŋ}_{ATP}$ to equal 34.6% for glucose [17]. The heat is unconverted into work, and energy dissipates from the body, leading to entropy generation. The heat transferred by glucose metabolism in the brain is described by Equation (4).
(4)$$Q_{entropy}=Q_{m}-W_{ATP}=\left(1-\eta \right)\;Q_{m}$$

The second law of thermodynamics was applied to the brain using Equation (5):(5)$$S_{gen}=\sum_{i=1}^{n}{\left(n\bar{s}\right)}_{i,p}-\sum_{i=1}^{m}{\left(n\bar{s}\right)}_{i,r}+\frac{-Q_{entropy}}{T_{env}}\;$$
where n and $\bar{s}$ indicate mole number and absolute entropy per unit mole of *i* component, respectively. 

The Gibbs free energy equation is a fundamental concept in thermodynamics that combines enthalpy and entropy changes. Its application is essential for understanding the spontaneity of chemical reactions and predicting their feasibility in different conditions [30]. The brain always aims to minimize the free energy change reflected in behavioural responses [31]. According to the hypothesis of Rietman et al. (2020) [32], Gibbs free energy increases, while seen in both early brain development and advanced age, are driven by different types of processes. In this study, the free energy of the brain was thermodynamically calculated based on Equation (6).
(6)$$∆G_{sys}=∆H_{sys}-∆TS_{sys}$$

Here, *G_sys_* denotes Gibbs free energy, *H_sys_* is the enthalpy, *T* is the absolute temperature, and *S_sys_* represents the entropy.

### 2.2. Grey Mouse Lemur Model

Pifferi et al. (2015) [26] aimed to observe the effect of dietary supplementation with n-3 PUFAs at an early age to prevent the impairment of brain glucose metabolism and glucose transport during ageing. Using the grey mouse lemur (*Microcebus murinus*) as an adult primate model, they performed a 12-month dietary intervention with two groups: a control group and the n-3 PUFA intervention group. The n-3 PUFA group received fish oil rich in EPA and DHA, while the control group was fed the same diet with an equivalent amount of olive oil. The daily intake of n-3 PUFA was approximately 6 mg EPA and 30 mg DHA per animal. The effect of n-3 PUFA supplementation on the cerebral metabolic rate of glucose (CMRGlc) was measured by positron emission tomography with ^18^F-fluorodeoxyglucose (^18^F-FDG-PET) in different brain regions. Details of the diets and the experimental methodologies are described in the original paper by Pifferi et al. (2015) [26]. The average data (mean) belonging to the control and intervention groups are presented in Table 2.

The % difference between the control and intervention groups was determined for each brain area, providing insights into the impact of long-term n-3 PUFA supplementation. The % difference was calculated using Equation (7).
(7)$$\%Difference=\frac{\left(Intervention\;value-Control\;value\right)}{Control\;value}\times 100$$

This formula allowed for the quantification of the relative change between the control and intervention groups in terms of entropy generation ($S_{gen})$ and Gibbs free energy ($G_{sys}$). 

### 2.3. Human Model

A study by Nugent et al. (2011) [27] investigated the effects of fish oil supplements (680 mg of DHA and 323 mg EPA/day) on brain glucose metabolism in healthy elderly individuals. Using ^18^F-FDG-PET, CMRGlc for the entire brain and various brain regions was measured in young (average 23 years) and healthy elderly (average 76 years) individuals who received n-3 PUFA supplementation for three weeks. The thermodynamic analysis used the collected data to assess the impact of fish oil supplementation on glucose metabolism and brain function, as detailed in Table 3. 

The difference between young and elderly participants before and after fish oil supplementation was calculated in terms of entropy generation ($S_{gen})$ and Gibbs free energy ($G_{sys}$) using Equation (8). The percentage of reduction between the difference between the young and elderly participants was determined using Equation (9).
(8)$$\%Difference=\frac{\left(After\;the\;intervention\;value-before\;the\;intervention\;value\right)}{Before\;the\;intervention\;value}\times 100$$
(9)$$\%Reduction=\frac{Difference\;before\;the\;intervention-Difference\;After\;the\;intervention}{Difference\;Before\;the\;intervention}\times 100$$

### 2.4. Statistical Analysis of Data 

The present study employed a thermodynamic analysis based on mean data extracted from published studies, and the analysis was carried out using the GetData Graph Digitizer. Due to the nature of the available data, a continuous data set for each participant was not directly accessible. Consequently, calculations of the entropy generation rate and Gibbs free energy were conducted utilizing mean values. While this approach precludes the possibility of performing statistical analyses at the individual level, it allows for comprehensive thermodynamic evaluation at the aggregate level. The mean data, derived from the synthesis of multiple studies, provided a basis for investigating thermodynamic trends across various brain areas. Additionally, this approach facilitates the identification of general patterns and tendencies in thermodynamic parameters, contributing to a broader understanding of the energetics within the studied contexts. It is important to note that the results presented herein reflect aggregated mean data, and any interpretations should consider the limitations associated with this approach. 

## 3. Results

Variations in entropy generation and Gibbs free energy were calculated by using previously published data of the cerebral metabolic rate of glucose (CMRGlc) in the whole brain and its regions as a function of n-3 PUFA supplementation (see Table 2 and Table 3). The results are presented in Table 4 and Table 5. 

### 3.1. Case Study 1: Grey Mouse Lemur Model

In the control group, the entropy generation rate ${(S}_{gen}$) varied among brain regions. The whole brain exhibited an entropy generation rate of 3.84 × 10^5^ kJ/100 g/K kg glucose per year. At the same time, specific regions such as the hippocampus, thalamus, and occipital lobe showed distinct values. Notably, the occipital lobe displayed the highest entropy generation rate among the studied areas (2.65 × 10^5^ kJ/100 g/K kg glucose per year). Upon intervention with n-3 PUFAs, there was a noteworthy increase in entropy generation rates across all brain regions. The hippocampus experienced the most substantial increment, with an entropy generation rate of 5.30 × 10^5^ kJ/100 g/K kg per year, marking a 68.2% difference from the control group. Conversely, the cerebellum showed the slightest difference, with 24.1%. 

The Gibbs free energy values were consistently negative, indicating changes in enthalpy and entropy that favoured a spontaneous process, as shown in Table 4 [30]. The evaluation of Gibbs free energy ($G_{sys})$ also demonstrated substantial alterations in the energy landscape of brain glucose metabolism (Table 2). In the control group, the whole brain exhibited a Gibbs free energy value of −6.47 × 10^4^ kJ/100 g/K kg glucose per year. Similar to entropy generation, specific brain regions showcased distinct values, with the occipital lobe having the lowest Gibbs free energy (−4.47 × 10^4^ kJ/100 g/K kg glucose per year). Following n-3 PUFA supplementation, there was an increase in Gibbs free energy across all brain regions. The hippocampus exhibited the most significant change, with a Gibbs free energy value of −8.93 × 10^4^ kJ/100 g/K kg per year, representing a 68.5% difference from the control group. The cerebellum exhibited the lowest difference, with 24.0% in comparison to the control group.

The differences in entropy generation and Gibbs free energy between the control and intervention groups underscore the impact of 12-month n-3 PUFA supplementation on the thermodynamic properties of brain glucose metabolism.

### 3.2. Case Study 2: Human Ageing Model

Table 5 presents the outcomes of entropy generation and Gibbs free energy for brain glucose metabolism for the human cohort detailed by Nugent et al. (2011) [27]. The findings reveal distinct patterns between age groups and the impact of 3-week fish oil supplementation.

The effect of short-term supplementation with n-3 PUFA for 3 weeks was not enough to provoke significant global thermodynamic changes. However, the thermodynamic assessment of brain glucose metabolism showed noteworthy differences in entropy generation between the young and elderly participants, as presented in Table 5. In the young group, the entropy generation rates varied across different brain regions. The whole brain exhibited an entropy generation rate of 1.41 × 10^4^ kJ/100 g/K kg glucose per 3w before the intervention, while the anterior cingulate demonstrated distinct values. After 3 weeks of n-3 PUFA supplementation, the whole brain entropy generation rate was 1.40 × 10^4^ kJ/100 g/K kg glucose per 3w, while the anterior cingulate entropy generation rate decreased by 10.42%. For the elderly group, there was a decrease in entropy generation rates across all brain regions except for the occipital lobe. The whole brain entropy generation rate decreased by 2.14%, while the anterior cingulate generated 16.52% more entropy after n-3 PUFA supplementation. When we compared the entropy generation rate for young and elderly participants for the specific brain regions, we found that supplementation with n-3 PUFA reduced the entropy generation difference between elderly and young individuals in the cerebellum (100% reduction) and specific parts of the brain cortex, namely the anterior cingulate (14.29% reduction), and occipital lobe (20% reduction).

The evaluation of Gibbs free energy also revealed significant alterations in the energy landscape of brain glucose metabolism (see Table 5). In the young group, the whole brain exhibited a Gibbs free energy value of −2.29 × 10^5^ kJ/100 g/K kg glucose per 3w before the intervention and a value of −2.28 × 10^5^ kJ/100 g/K kg per 3w after n-3 PUFA supplementation. Specifically, the anterior cingulate, posterior cingulate, and frontal lobe brain regions displayed distinct Gibbs free energy value increases of 10.26%, 3.38%, and 4.44% following the intervention. The highest difference (10.42% decrease) was observed in the anterior cingulate, and the lowest difference was 0.44% (increase), observed in the whole brain after 3w n-3 PUFA supplementation. Following n-3 PUFA supplementation, there was an increase in Gibbs free energy across all brain regions except the anterior cingulate in the elderly adults. The anterior cingulate showed the most significant change, with a Gibbs free energy value of −2.17 × 10^5^ kJ/100 g/K kg glucose per 3w, representing a 16.04% difference from before the intervention. Moreover, with n-3 PUFA supplementation, Gibbs the free energy results showed a reduction in the differences between elderly and young individuals only for the anterior cingulate (60.64%).

## 4. Discussion

A growing body of evidence indicates that assessing thermodynamic properties can determine living organisms’ disorder levels and lifespan [29,33,34]. The changes in thermodynamic properties with age draw attention to the need to establish effective strategies to support brain energy supply during ageing. The brain is a highly complex and metabolically active organ. A decrease in metabolism may cause a reduction in the entropy generation rate and an increase in Gibbs free energy. The results found in the human cohort of this study demonstrated that the rates of entropy generation in the brain and its regions decline as individuals age. This agrees with data recently reported in a cohort of 20- to 82-year-old cognitively normal adults, which illustrated that the ageing process is associated with a decrease in entropy generation rates in the whole brain and its regions [13]. Glucose is a vital fuel for brain energy metabolism, and measuring the thermodynamics of regional CMRGlc enables us to obtain information about regional brain activity. It has clinical importance in assessing the normal function of the brain areas and pathological alterations [35]. An imbalance in oxidative and energy metabolism may cause glucose hypometabolism. This is associated with several diseases, such as Alzheimer’s disease (AD), a major depressive disorder; Huntington’s disease; and epilepsy-related diseases [36,37]. In healthy ageing, decreases in brain functionality related to energy metabolism-related pathways are observed. Based on epidemiological, experimental, and clinical research, the preclinical stages of neural disorders such as AD and Parkinson’s disease are considered to impair neural bioenergetics in normal ageing [1].

In this field, the diet dependence of the entropy generation and accumulation rates of a living system is emerging as a promising tool for developing personalized nutrition strategies [29,35,38]. The grey mouse lemur model results showed that long-term n-3 PUFA supplementation increased the entropy generation rate and decreased Gibbs free energy in the whole brain and specific brain regions. Such an increase in entropy, along with the decrease in Gibbs free energy, indicates a glucose metabolism process that is favourable in terms of bioenergetics, capable of releasing energy, and a facilitator of reactions that occur spontaneously [39,40]. These changes in thermodynamic properties helped establish the role of n-3 PUFA in regulating brain glucose metabolism, as described by Pifferi et al. (2015) [26]. The incorporation of marine oil (daily intake of about 6 mg EPA and 30 mg DHA) into participants’ diet affected the entropy and Gibbs free energy of glucose metabolism after long-term supplementation in healthy young animals, likely before the impairment in glucose metabolism associated with age. The effects of n-3 PUFA intake as a function of age have been demonstrated in neurotransmission processes, which demand high levels of brain ATP [25]. Data from the lemur model indicated that supplementation with n-3 PUFA resulted in greater effects on entropy generation rates and Gibbs energy in the hippocampus and occipital lobe than in the other brain regions. Both the hippocampus and occipital lobe are involved in memory formation (further details on the functions of the hippocampus and occipital lobe are defined in Appendix A, Table A1). A pronounced age-derived effect on the functional connectivity networks within brain occipital areas was previously described in healthy young and elderly subjects during episodic and working memory demand [41]. The young subjects showed highly symmetrical neural networks, accompanied by a strong coupling between parietal and occipital regions. In contrast, a substantial left-hemispheric asymmetry with decreased connectivity within occipital areas was described for the healthy seniors. Indeed, the hippocampus is the primary target and most severely affected region in several neuropsychiatric disorders, such as Alzheimer’s disease (AD), epilepsy, etc. [42]. On the other hand, the cerebellum is only partially targeted, resulting in minor effects caused by n-3 PUFA supplementation. Several studies on post-mortem tissues suggest that the cerebellum is resistant to ageing when comparing the whole brain and other brain regions. This may result in the cerebellar leukocyte telomere length not changing with age, and mitochondrial DNA deletions and oxidative damage are rarer in the cerebellum. Moreover, the weakest alterations in glucose metabolism occur within the cerebellum [43]. The difference between the control and intervention groups in terms of entropy generation and Gibbs free energy indicates that n-3 PUFA supplementation improved glucose metabolism efficiency, potentially promoting cellular functions and overall brain health.

Supplementation with n-3 PUFA did not affect entropy generation rates and Gibbs free energy associated with brain glucose metabolism in the human cohort. Short-term supplementation with n-3 PUFA for 3 weeks did not address significant global thermodynamic changes. Still, it seemed to prevent, to some extent, the hypometabolism in the cerebellum and specific parts of the brain cortex, namely the anterior cingulate and the occipital lobe. These brain cortex regions are related to motor control, as is the cerebellum, as well as certain rational cognitive functions, such as partial processing, the recognition and appreciation of what is being seen, the perception of distance and depth, and the initiation of reflex actions, which play key parts in language- and emotion-related learning. However, n-3 supplementation reduced the difference between the young and elderly subjects in terms of Gibbs free energy and entropy generation. This means that n-3 supplementation helped improve glucose metabolism efficiency in these areas. Since not all brain regions were affected, in this case, the elderly subjects may have had a healthy brain status and not yet experienced a decline in brain function. Nugent et al. (2011) [27] have already described that the absence of an apparent effect of n-3 PUFA supplementation on glucose metabolism in the brain could be attributed to the short n-3 PUFA dietary supplementation period. It is important to note that this human study had limitations, including a small number of subjects and the fact that the study’s calculations were based on global data on glucose metabolism. Moreover, we did not have any information related to brain oxygen consumption (CMRO_2_), aerobic glycolysis (AG), and cerebral blood flow (CBF). Therefore, we assumed that glucose metabolism was dependent solely on oxidative phosphorylation.

The daily amount of EPA and DHA consumed in the human model study was 323 mg EPA and 680 DHA. When we compared the human model with the grey mouse lemur model, the daily intake of EPA and DHA stayed under the amounts consumed by the intervention group in the grey mouse lemur model. Moreover, daily DHA+EPA consumption by young and elderly groups was also below the upper recommendation of 2000 mg/d EPA+DHA intake [44]. The brain structure of the grey mouse lemur is similar to that of the human brain. Therefore, the grey mouse lemur, one of the most convenient organisms for brain research, has been proposed as a model organism to understand the mechanisms underlying age-associated neurodegenerative diseases [31,45]. Studies suggest that it enables the valuable translation of results to human clinical research [46]. Therefore, we used the grey mouse lemur CMRGlc data from Pifferi et al. (2015) [26]. They adjusted the intake of EPA (20:5 n-3) and DHA (22:6 n-3), which constituted approximately 0.06% and 0.3% of the total energy intake, respectively. These amounts correspond to 502 mg EPA and 2510 mg DHA when considering an average healthy adult’s daily intake of 8368 kJ. This amount is higher than the upper limit recommended for EPA+DHA intake, which is 2000 mg per day [44].

The present grey mouse lemur model data indicate that longer-term n-3 PUFA supplementation corresponds to 3012 mg/d of EPA+DHA for an adult human, resulting in increased entropy generation and decreased Gibbs energy, which is associated with increased brain glucose uptake and metabolism. The main novelty of this paper is the observation that the entropy generation rate and Gibbs energy rate, both associated with brain glucose metabolism, are affected during long-term dietary n-3 PUFA supplementation, independently of the brain regions we considered. As previously noted, FDG-PET imaging is a valuable technique for evaluating the brain’s overall health status via kinetic imaging analysis to determine CMRGlc. However, studies using this technique are remarkably lacking due to the hazardous effects of radiation on humans. We suggest that specific groups requiring FDG-PET imaging should be used in future studies to evaluate the impact of marine products or n-3 PUFA supplements on the brain, and for comparisons, the data from the literature may be helpful. On the other hand, it is crucial to adjust the daily amount of the supplement being consumed and its consumption period. The grey mouse lemur model indicates that long-term (12 months) fish oil supplementation may improve brain function by increasing CMRGlc. The human model suggests that short-term fish oil supplementation does not significantly affect brain glucose metabolism. Based on these observations, we suggest that studies investigating the effects of marine products or n-3 PUFA supplements on brain activity following a long-term supplementation period should be designed. Another critical point is to know about participants’ diets before the study. With this knowledge, it is possible to assess the inevitable effects of marine products on the brain.

This study presents the advantage of using thermodynamic properties to understand how marine products or n-3 PUFA supplements influence the human brain. All contemporary biological ageing theories share a common foundation rooted in alterations in molecular structure and consequent functional changes due to entropy-related processes [12]. Adopting a thermodynamic perspective elucidates the advantages of maintaining good health.

### Limitations of the Study

While our study provides valuable insights into the thermodynamic assessment of the impacts of n-3 PUFA supplementation on ageing brain functions, it is important to admit that this study had certain limitations that may affect the explication of our findings. One of these significant limitations was the lack of a data set to assess data in terms of thermodynamics. The present study employed a thermodynamic analysis based on mean CMRGlc data obtained from published studies. Due to the nature of the available data, a continuous data set for each participant was not directly accessible. This limitation meant that our results relied on aggregated data rather than individual-level data, which may have restricted the depth of our analysis and statistical validation. Consequently, calculations of entropy generation and Gibbs free energy were conducted utilizing mean values. Although this approach enables comprehensive thermodynamic assessment at the aggregate level, there is an absence of individual-level statistical analyses, which limited our capability to evaluate variability and make precise inferences about individual differences.

To address these limitations and improve the transparency and reproducibility of research results, supporting data-sharing practices within the scientific community is necessary. Encouraging researchers to share raw data sets at an individual level, along with detailed methodological information, could help independent researchers to conduct secondary analysis and validate their findings. Furthermore, the establishment of standardized protocols for data storage and sharing, along with incentives and recognition for data-sharing efforts, could foster a culture of openness and collaboration in scientific research. By strengthening data-sharing practices, future research can overcome the limitations associated with aggregated data and contribute to a more comprehensive understanding of brain energetics and its influences on health and disease.

## 5. Conclusions

In conclusion, our study emphasizes the pivotal role of thermodynamics, specifically Gibbs free energy and entropy generation, in unravelling the intricate dynamics of brain ageing and health. Evaluating the thermodynamic properties of glucose metabolism offers a unique perspective on age-related modifications, providing valuable insights into living organisms’ disorder levels and lifespans.

The impact of fish oil supplementation and n-3 polyunsaturated fatty acids (PUFAs) on brain health is evident. Long-term n-3 PUFA supplementation induces favourable thermodynamic changes in the brain as a whole and in critical regions like the hippocampus and occipital lobe, which are crucial for memory formation. The entropy generation rate difference between the control and invention groups was highest in the hippocampus, with a difference of 68.2%. At the same time, notably, the cerebellum, which is resistant to ageing, remained relatively unaffected, showing the slightest difference in the cerebellum (24.1%). Short-term n-3 PUFA supplementation exhibits resilience against hypometabolism in select brain regions. The difference in entropy generation rate between young and elderly individuals after n-3 PUFA supplementation decreased in the cerebellum (100% reduction), anterior cingulate (14.29% reduction), and occipital lobe (20% reduction). Our results highlight nuanced effects on motor control and particular cognitive functions, including emotion-related learning, social cognition, spatial processing, and recognizing and appreciating what is being seen.

Recognizing the significance of thermodynamics in nutritional interventions provides a pathway for personalized strategies to maintain optimal brain health, particularly in ageing and neurodegenerative disorders. This study advances our understanding of ageing brain metabolic changes. It underscores the need for comprehensive studies to grasp the full impact of dietary interventions on brain activity in neuroscience.

## Figures and Tables

**Figure 1 nutrients-16-00631-f001:**
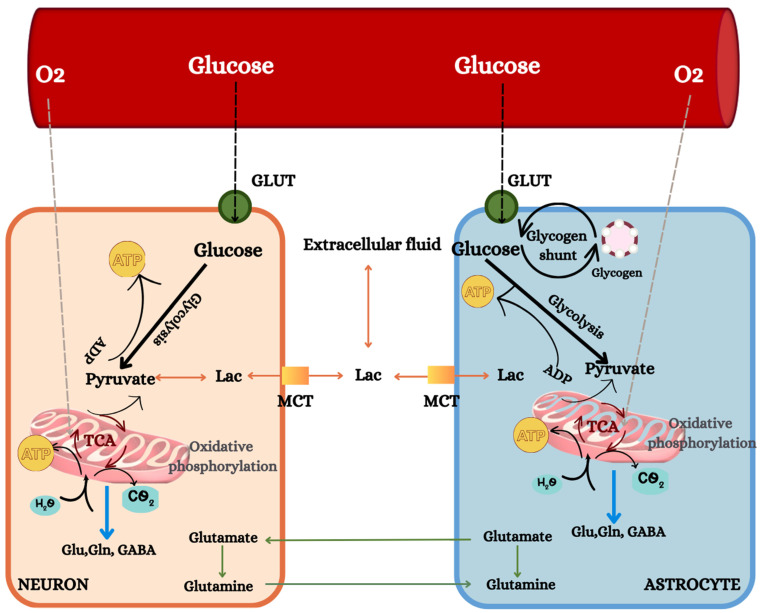
Glucose transporters (GLUTs) play a crucial role in facilitating the transportation of glucose from capillaries to neurons and astrocytes. Once inside the cells, the glycolytic pathway undergoes breakdown through the glycolytic pathway, forming glucose into pyruvate and yielding a net of two ATP molecules per glucose molecule. Further metabolic processes take place within the mitochondria, where pyruvate is subjected to oxidative metabolism via the tricarboxylic acid (TCA) cycle. The TCA cycle generates approximately 30 or 36 ATP molecules. The glyceraldehyde-3-phosphate dehydrogenase reaction catalyses a pivotal step in converting glucose to pyruvate. This reaction involves the regeneration of NAD+ from NADH, a coenzyme produced during the reaction. The malate aspartate shuttle (MAS) facilitates the transfer of cytoplasmic NADH to the mitochondria, where it is oxidized through the electron transport chain (ETC). However, under specific conditions such as hypoxia or anoxia, pyruvate can be converted to lactate through the lactate dehydrogenase (LDH) reaction. Lactate accumulation inside cells triggers a reversal of the LDH reaction. Monocarboxylic acid transporters (MCT) come into play to release lactate from the cell. This process eliminates pyruvate as an oxidizable substrate for the cell and limits the ATP yield per glucose to two. In the excitatory neuronal activity, glutamate (Glu) is released, and a significant portion is absorbed through GLT-1 in astrocytes. This activation of GLT-1 stimulates Na+/K+-ATPase, subsequently triggering aerobic glycolysis. The metabolic analogue of glucose, 2-deoxyglucose (2-DG), is metabolized by hexokinase, similar to its ^18^F analogue used in PET. However, further metabolism is halted at this stage, resulting in the trapping of radioactivity in the form of 2-DG-6-phosphate (2-DG-6P), the signal detected by PET. Adapted from [4,5,6,7].

**Figure 2 nutrients-16-00631-f002:**
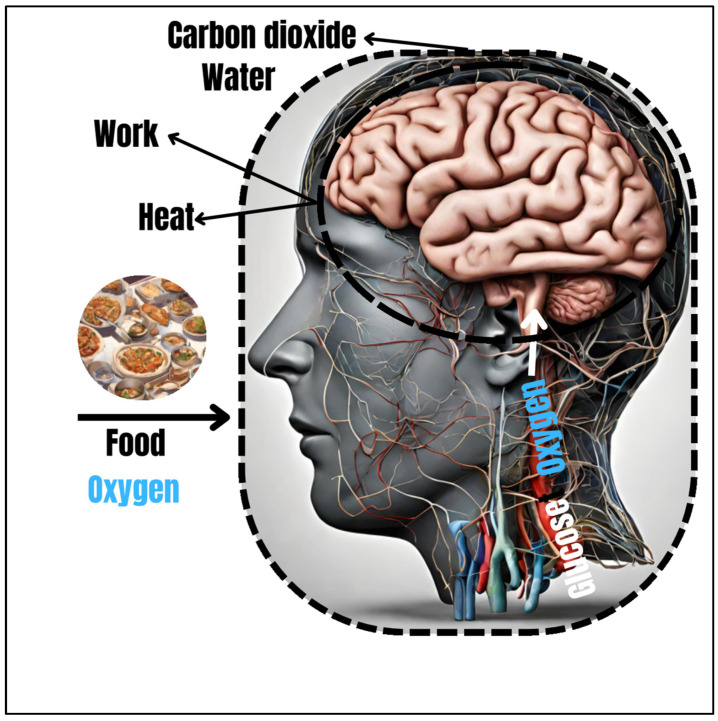
Graphical demonstration of a healthy brain as an open thermodynamic system.

**Table 1 nutrients-16-00631-t001:** Thermodynamic properties of the chemicals (data adapted from Kuddusi, 2015 [29]).

Chemical	$h_{f}^{-o}$ (kJ/kmol)	$h_{298K}^{-}$ (kJ/kmol)	$h_{310K}^{-o}$ (kJ/kmol)	$s_{i}$ * (kJ/kmol K)
Glucose C_6_H_12_O_6_	−1.260 × 10^3^	-	-	212 (at 298 K)
O_2_	0	8682	9030	218.02 (at 298 K) 219.68 (at 310 K)
H_2_O	−241,820	9904	10,302	218.9 (at 310 K)
CO_2,_	−393,520	9364	9807	243.64 (at 310 K)
N_2_	0	8669	9014	193.66 (at 310 K)

* $s_{i}$ denotes the absolute entropy for each component per kmol at a specific temperature (T (K) and environment pressure (1 atm), as detailed by Kuddusi (2015) [29].

**Table 2 nutrients-16-00631-t002:** The average CMRGlc for control and n-3 LCPUFA-supplemented groups of grey mouse lemurs (data obtained from Pifferi et al., 2015 [26]).

Brain Area	Control CMRGLc (µmol/100 g/min)	Intervention CMRGLc (µmol/100 g/min)
Whole brain	13.9	20.5
Hippocampus	11.4	19.2
Thalamus	16.8	25.5
Cerebellum	19.4	24.1
Caudate nucleus	12.8	20.0
Temporal lobe	12.1	17.7
Occipital lobe	9.6	17.1
Frontal lobe	14.0	20.8

**Table 3 nutrients-16-00631-t003:** The average CMRGlc for healthy young and elderly individuals before and after n-3 PUFA supplementation. (data obtained from Nugent et al., 2010 [27]).

	Before		After	
Brain Area	Young CMRGLc (µmol/100 g/min)	Elderly CMRGLc (µmol/100 g/min)	Young CMRGlc (µmol/100 g/min)	Elderly CMRGlc (µmol/100 g/min)
Cerebellum	42.7	42.3	42.6	42.6
Hippocampus	38.6	37.0	38.0	35.5
Anterior cingulate	50.4	40.2	45.1	46.7
Posterior cingulate	57.3	54.5	55.4	48.2
Frontal lobe	53.3	50.4	51.0	49.5
Parietal lobe	49.5	49.2	49.2	48.3
Occipital lobe	50.4	53.5	52.0	50.1
Temporal lobe	47.6	46.1	47.0	45.4
Whole brain	49.2	48.9	48.9	47.9

**Table 4 nutrients-16-00631-t004:** Entropy generation and Gibbs free energy values for the control and n-3 PUFA-supplemented groups of the grey mouse lemur model.

	**Entropy Generation**		
**Brain Area**	**Control** $S_{gen}$ **(kJ/100 g/K kg Glucose per Year)**	**Intervention** $S_{gen}$ **(kJ/100 g/K kg per Year)**	**Difference (%)**
Whole brain	3.84 × 10^5^	5.66 × 10^5^	47.4
Hippocampus	3.15 × 10^5^	5.30 × 10^5^	68.2
Thalamus	4.64 × 10^5^	7.04 × 10^5^	51.7
Cerebellum	5.36 × 10^5^	6.65 × 10^5^	24.1
Caudate nucleus	3.53 × 10^5^	5.52 × 10^5^	56.2
Temporal lobe	3.34 × 10^5^	4.89 × 10^5^	46.4
Occipital lobe	2.65 × 10^5^	4.47 × 10^5^	68.7
Frontal lobe	3.87 × 10^5^	5.74 × 10^5^	48.3
	**Gibbs Free Energy**		
**Brain Area**	**Control** $G_{sys}$ **(kJ/100 g/K kg Glucose per Year)**	**Intervention** $G_{sys}$ **(kJ/100 g/K kg per Year)**	**Difference (%)**
Whole brain	−6.47 × 10^4^	−9.54 × 10^4^	47.4
Hippocampus	−5.30 × 10^4^	−8.93 × 10^4^	68.5
Thalamus	−7.82 × 10^4^	−1.19 × 10^5^	52.2
Cerebellum	−9.03 × 10^4^	−1.12 × 10^5^	24.0
Caudate nucleus	−5.96 × 10^4^	−9.30 × 10^4^	56.0
Temporal lobe	−5.63 × 10^4^	−8.23 × 10^4^	46.2
Occipital lobe	−4.47 × 10^4^	−7.54 × 10^4^	68.7
Frontal lobe	−2.21 × 10^5^	−2.14 × 10^5^	48.7

**Table 5 nutrients-16-00631-t005:** Entropy generation and Gibbs free energy values for healthy young and elderly individuals before and after fish oil supplementation.

	**Entropy Generation**			
	**Before**		**After**	
**Brain Area**	**Young** $S_{gen}$ **(kJ/100 g/K kg Glucose per 3w)**	**Elderly** $S_{gen}$ **(kJ/100 g/K kg per 3w)**	**Young** $S_{gen}$ **(kJ/100 g/K kg per Glucose 3w)**	**Elderly** $S_{gen}$ **(kJ/100 g/K kg per Glucose 3w)**
Cerebellum	1.22 × 10^7^	1.21 × 10^7^	1.22 × 10^7^	1.22 × 10^7^
Hippocampus	1.10 × 10^7^	1.06 × 10^7^	1.09 × 10^7^	1.01 × 10^7^
Anterior cingulate	1.44 × 10^7^	1.15 × 10^7^	1.29 × 10^7^	1.34 × 10^7^
Posterior cingulate	1.64 × 10^7^	1.56 × 10^7^	1.58 × 10^7^	1.38 × 10^7^
Frontal lobe	1.52 × 10^7^	1.44 × 10^7^	1.46 × 10^7^	1.42 × 10^7^
Parietal lobe	1.42 × 10^7^	1.41 × 10^7^	1.41 × 10^7^	1.38 × 10^7^
Occipital lobe	1.44 × 10^7^	1.53 × 10^7^	1.49 × 10^7^	1.43 × 10^7^
Temporal lobe	1.36 × 10^7^	1.32 × 10^7^	1.34 × 10^7^	1.30 × 10^7^
Whole brain	1.41 × 10^4^	1.40 × 10^4^	1.40 × 10^4^	1.37 × 10^4^
	**Gibbs Free Energy**			
	**Before**		**After**	
**Brain Area**	**Young** $G_{sys}$ **(kJ/100 g/K kg Glucose per 3w)**	**Elderly** $G_{sys}$ **(kJ/100 g/K kg per 3w)**	**Young** $G_{sys}$ **(kJ/100 g/K kg per glucose 3w)**	**Elderly** $G_{sys}$ **(kJ/100 g/K kg per glucose 3w)**
Cerebellum	−1.99 × 10^5^	−1.97 × 10^5^	−1.98 × 10^5^	−1.98 × 10^5^
Hippocampus	−1.80 × 10^5^	−1.72 × 10^5^	−1.77 × 10^5^	−1.65 × 10^5^
Anterior cingulate	−2.34 × 10^5^	−1.87 × 10^5^	−2.10 × 10^5^	−2.17 × 10^5^
Posterior cingulate	−2.67 × 10^5^	−2.54 × 10^5^	−2.58 × 10^5^	−2.24 × 10^5^
Frontal lobe	−2.48 × 10^5^	−2.34 × 10^5^	−2.37 × 10^5^	−2.30 × 10^5^
Parietal lobe	−2.30 × 10^5^	−2.29 × 10^5^	−2.29 × 10^5^	−2.25 × 10^5^
Occipital lobe	−2.34 × 10^5^	−2.49 × 10^5^	−2.42 × 10^5^	−2.33 × 10^5^
Temporal lobe	−2.21 × 10^5^	−2.14 × 10^5^	−2.19 × 10^5^	−2.11 × 10^5^
Whole brain	−2.29 × 10^5^	−2.28 × 10^5^	−2.28 × 10^5^	−2.23 × 10^5^

## Data Availability

Data are contained within the article.

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
