# Peer review of "Thermodynamic Analysis to Evaluate the Effect of Diet on Brain Glucose Metabolism: The Case of Fish Oil"

_nutrients, 2024, doi:10.3390/nu16050631_

Round 1

Reviewer 1 Report

Comments and Suggestions for Authors

Table 1: explain the symbol si

Supplementation of n-3 PUFA: different ratios of EPA:DHA were fed to the mice model and the human model. Can the results be used for comparison or eg the results from the mice study can be used to explain the results for the human model?

Reference [15]: Actually the researchers that calculated this can be found in the following reference https://doi.org/10.1155/2009/186723

Results: 3.2: Gibb's energy seems to be quite close for the two groups. So how is the statement that "healthy-aged adults tend to accumulate less entropy and Gibbs energy" justified?

Results: Table 5. Please explain how you calculate the %reduction and whether these are considered significant. In % they seem high, but in absolute values seem non significant. So, how important is the physiological effect?

Reviewer 2 Report

Comments and Suggestions for Authors

The authors of this manuscript describe the application of thermodynamics to assess the effect of omega 3 fish oil supplementation on glucose metabolism within the brain. They use previously published cerebral glucose metabolic rates measured by 18F-FDG-PET in a grey mouse lemurs and humans.

The manuscript has some significant issues that make it unsuitable for publication. These are outlined below:

Major issues:

My primary concern with this manuscript is the lack of statistical analysis. A "lowest p value" is mentioned once in the results section on line 261, but no values are actually provided. Further, there is no mention of statistical methods in the manuscript.

The data that is provided in the Results (in Tables 4 and 5) do not indicate whether values are means or medians of entropy generation and Gibbs free energy. Additionally, no information about data spread (standard deviation or interquartile range) is given. This makes assessment of the data impossible.

Minor issues:

The abstract at 269 words exceeds the maximal length specified in the instructions to authors (200 words). This should be shortened.

Sections of the paper should be rewritten so that appropriate content is found in the appropriate section. For instance, the Results section contains references and has interpretation results, these should be moved to the Methods or Discussion sections.

Comments on the Quality of English Language

Minor editing of English required.

Round 2

Reviewer 2 Report

Comments and Suggestions for Authors

I appreciate the author's clarification of the methods and the reason for a lack of statistical analyses. The addition of the section describing this in the methods has addressed my concern. 

Overall, I feel that the authors have sufficiently addressed my concerns.

I do have one recommendation. I feel that the discussion section could benefit from the addition of a paragraph on the limitations of the study, where this lack of statistical analyses could be addressed further. Perhaps adding suggestions of how data sharing could be improved to allow continuous data sets for each participant to be available for analyses such as those in this study.

Author Response

Dear Reviewer,

Thank you for your positive feedback on the revised manuscript and the constructive recommendation, we appreciate your thorough review and are glad to hear our revisions have sufficiently addressed your concerns.

Regarding your recommendation for further improvement in the discussion section, we agree that it would be valuable to include a part of the limitation of the study. Therefore, we added the limitations of the study into the Discussions on page 14, paragraph 4 and line 460 and specifically addressed the lack of statistical analysis due to the absence of a raw data set, and provided insights into the importance of data sharing and how data sharing improves future research in this area.

Limitations of the study

While our study provides precious insights into the thermodynamic assessment of the impacts of n-3 PUFA supplementation on aging brain functions, it is important to admit the certain limitations that may affect the explication of our findings. One of the significant limitations was the lack of a data set to assess data in terms of thermodynamics. The present study employed thermodynamic analysis based on mean CMRGlc data obtained from published studies. Due to the nature of the available data, a continuous data set for each participant was not directly accessible. This limitation leads our results to rely on aggregated data rather than individual-level data, which may restrict the depth of analysis and statistical validation. Consequently, calculations of entropy generation and Gibb’s free energy were conducted utilizing mean values. Although this approach enables comprehensive thermodynamic assessment at the aggregate level, there is an absence of individual-level statistical analyses that limit the capability of evaluating variability and making precise inferences about individual differences.

To address these limitations and improve the transparency and reproducibility of research results, supporting data-sharing practices within the scientific community is necessary. Promoting researchers to share raw data sets at an individual level, along with detailed methodological information, can facilitate independent researchers to secondary analysis and validate their findings. Furthermore, the establishment of standardized protocols for data storage and sharing, along with incentives and recognition for data-sharing efforts, could foster a culture of openness and collaboration in scientific research. By strengthening data-sharing practices, future research can overcome the limitations associated with aggregated data and contribute to a more comprehensive understanding of brain energetics and its influences on health and disease. “